# A Genomic Analysis of the *Bacillus* Bacteriophage *Kirovirus kirovense* Kirov and Its Ability to Preserve Milk

**DOI:** 10.3390/ijms241612584

**Published:** 2023-08-09

**Authors:** Olesya A. Kazantseva, Anna V. Skorynina, Emma G. Piligrimova, Natalya A. Ryabova, Andrey M. Shadrin

**Affiliations:** 1Laboratory of Bacteriophage Biology, G.K. Skryabin Institute of Biochemistry and Physiology of Microorganisms, Pushchino Scientific Center for Biological Research of the Russian Academy of Sciences, Federal Research Center, Prospect Nauki, 5, 142290 Pushchino, Russia; s_an.net@mail.ru (A.V.S.); e.piligrimova@ibpm.ru (E.G.P.); ryabova@phys.protres.ru (N.A.R.); 2Institute of Protein Research RAS, Institutskaya St., 4, 142290 Pushchino, Russia

**Keywords:** *Bacillus cereus*, *Bacillus thuringiensis*, bacteriophage, phage, *Caudoviricetes*, *Andregratiavirinae*, *Kirovirus*, siphovirus, biocontrol, food preservation, milk decontamination, genome sequencing

## Abstract

Bacteriophages are widely recognized as alternatives to traditional antibiotics commonly used in the treatment of bacterial infection diseases and in the food industry, as phages offer a potential solution in combating multidrug-resistant bacterial pathogens. In this study, we describe a novel bacteriophage, *Kirovirus kirovense* Kirov, which infects members of the *Bacillus cereus* group. *Kirovirus kirovense* Kirov is a broad-host-range phage belonging to the *Caudoviricetes* class. Its chromosome is a linear 165,667 bp double-stranded DNA molecule that contains two short, direct terminal repeats, each 284 bp long. According to bioinformatics predictions, the genomic DNA contains 275 protein-coding genes and 5 tRNA genes. A comparative genomic analysis suggests that *Kirovirus kirovense* Kirov is a novel species within the *Kirovirus* genus, belonging to the *Andregratiavirinae* subfamily. *Kirovirus kirovense* Kirov demonstrates the ability to preserve and decontaminate *B. cereus* from cow milk when present in milk at a concentration of 10^4^ PFU/mL. After 4 h of incubation with the phage, the bacterial titer drops from 10^5^ to less than 10^2^ CFU/mL.

## 1. Introduction

*Bacillus cereus sensu lato* (*B. cereus s.l.*) is a group of ubiquitous soil bacteria. In soil samples, the concentration of these bacteria can exceed 10^5^ colony-forming units per mL (CFU/mL) [1]. Some strains of this group of bacteria are pathogenic for mammals and are capable of synthesizing a number of toxins [2]. Members of the *B. cereus* group can grow even at food storage temperatures [1,3]. For this reason, *B. cereus s.l.* is among the leading bacterial species that causes food poisoning outbreaks [1] and in certain severe cases, it can lead to endocarditis [4], endophthalmitis [5], and even death [6,7,8]. Regarding food products, *B. cereus s.l.* is often found in rice-based products [9], as well as milk and its derivatives [10,11,12].

Bacteriophages are natural enemies of bacteria and have the ability to dismantle biofilms [13,14]. Bacterial death caused by phage-induced lysis is pervasive in nature. It is estimated that around 20% of the bacterial population is eliminated daily due to bacteriophage lysis [15]. The incidence of infections associated with antibiotic-resistant bacterial strains is increasing every year [16]. Routine approaches for preventing food contamination and treating bacterial infections in vitro primarily rely on the use of antibiotics. However, the effectiveness of these approaches is often limited due to the toxic effects of antibiotics and the development of bacterial resistance. One effective alternative strategy for controlling potentially pathogenic species involves the use of bacteriophages as biocontrol agents [17]. The application of bacteriophages enables the selective targeting and elimination of harmful bacteria, while minimizing the risks associated with antibiotic toxicity and the development of bacterial antibiotic resistance. This approach appears to be a promising alternative to traditional antibacterial methods. For instance, in 2007, Intralytix, Inc. received approval from the Food and Drug Administration to use the bacteriophage cocktail called ListShield™ for controlling the foodborne bacterial pathogen *Listeria monocytogenes* [18]. Further, Intralytix, Inc. expanded its range of preparations for the biocontrol of food and animal feed, targeting pathogens such as *E. coli* [19], *Salmonella* [20], and others.

An impressive variety of the bacteriophage-infecting *Bacillus cereus* group has already been isolated and taxonomically classified. Importantly, some of the phages have been shown to infect pathogenic *B. cereus sensu stricto* and *B. anthracis* strains [14,21]. Although only virulent phages are considered safe enough to be applied in disease treatment, temperate phages can also be useful; they possess lytic enzymes that can be obtained in a recombinant form and used independently from the phages [22]. Thus, bacteriophages and their lytic enzymes are gaining more recognition as promising pathogen control agents that can be applied in disease therapy and food industry together with (or instead of) classical antibiotics and antimicrobial agents.

In this study, we isolate and characterize a new bacteriophage named *Kirovirus kirovense* Kirov (Kirov for short) preying on *B. cereus s.l.* which may be effective against *B. cereus* group pathogens. Additionally, we assess the ability of the Kirov bacteriophage to eliminate *B. cereus s.l.* contamination in cow’s milk.

## 2. Results

### 2.1. Phage Isolation, Host Range, Plaque Morphology, and Transmission Electron Microscopy

The Kirov bacteriophage was isolated from a soil sample collected in Kirov, Kirov region, Russian Federation, in 2017. According to the International Code of Virus Classification and Nomenclature (ICVCN) [23], the phage was named in the binomial format consisting of a Latinized genus name and the species epithet based on geographical location: *Kirovirus kirovense*, phage strain Kirov (hereafter referred to as Kirov). The strain *B. tropicus* ATCC 4342, sensitive to Kirov phage, was used as a propagation strain. The host range analysis showed that Kirov was capable of infecting 12 (out of 42) *Bacillus* strains, whereas 25 strains of the *Bacillus cereus* group and 5 other *Bacillus* strains were resistant to the phage infection (Appendix A). On the lawn of the host strain *B. tropicus* ATCC 4342, Kirov produced clear plaques with a diameter of 3–6 mm (Figure 1a,b).

A transmission electron microscopy (TEM) analysis revealed that the phage possesses an icosahedral non-elongated head of approximately 78.3 ± 2.1 nm in diameter attached to a characteristic long non-contractile flexible tail of approximately 437.6 ± 12.1 nm in length (not including the baseplate structure) and 12.3 ± 0.9 nm in width. The tale ends with a baseplate with a protruding fiber-like structure (Figure 1c). The micrograph shows that some phage virions interact with each other by their tail fibers (Appendix A). The tail contains about 110 visible striations (disk-like structures). The phage has all the characteristic morphological features of the siphovirus morphotype (Figure 1c).

### 2.2. Genome Characteristics

Phage DNA was extracted and sequenced on the Illumina platform, resulting in a single circular contig with the length of 165,383 base pairs (bp), an average coverage of 335× and a GC content of 35.5% from the sequencing reads. Genome ends were verified by restriction analysis (Figure 2a) and the RAGE method, followed by Sanger sequencing (Figure 2b). The analyses revealed the presence of two short direct terminal repeats (DTRs) at the ends of the chromosome, each 284 bp in length. Therefore, the length of the Kirov chromosome was 165,667 bp. A schematic representation of the phage chromosome is shown in Figure 2c.

The genome of Kirov contains 280 predicted genes, including 275 protein-coding genes and 5 tRNA genes (Appendix A, respectively). Out of the 275 protein-coding genes, 108 (39.3%) were functionally assigned. Protein-coding genes were classified into several functional modules (Figure 3) based on the predicted functions. DNA packaging genes encode small and large subunits of terminase similar to those of Izhevsk [24] and PBC2 [25] phages. The structural genes identified encode proteins that make up mature viral particles, as well as those involved in virion assembly: portal protein, putative prohead protease, major capsid protein, head completion protein, putative tail terminator protein, putative tail assembly chaperone, tail tube protein, tail completion protein, tape measure protein, distal tail protein, tail endopeptidase, receptor-binding protein, endosialidase, and putative tail fiber chaperone. The tail genes are typical of phages with the siphovirus morphotype that infect Gram-positive hosts [26]. Lytic genes encode one N-acetylmuramoyl-L-alanine amidase and two holin-like proteins. N-acetylmuramoyl-L-alanine amidase (ORF 56 product) is 89% identical to the endolysin (locus tag X915_gp217) of the giant *Bacillus*-infecting phage vB_BanS-Tsamsa (a siphovirus effective against several *B. anthracis* strains) [27] and 88% identical to the thermostable amidase Ply57 (locus tag Izhevsk_197) of the *Bacillus* phage Izhevsk [24].

Replication-, reparation-, and recombination-related proteins of the Kirov phage include replication initiator, RecD-like DNA helicase, RuvA C-terminal domain-containing protein, single-stranded (ss) DNA-binding protein, replicative DNA helicase (DnaB), DNA primase (DnaG), RecJ-type ssDNA-specific exonuclease, subunits A and B of DNA gyrase/topoisomerase IV, DNA ligase, DNA polymerase III alpha subunit, and RuvC-like Holliday junction resolvase.

Kirov encodes a large set of proteins involved in nucleic acid metabolism, including SAM-dependent methyltransferase, thymidylate kinase, putative nucleotidyltransferases, alpha and beta subunits of ribonucleotide-diphosphate reductase, thioredoxin, dUTPase, guanylate kinase, RNA ligase, polynucleotide kinase, 5’-nucleotidase, thymidylate synthase, dihydrofolate reductase, thymidine kinase, and RNA helicase-ribonuclease.

Additionally, the phage possesses three genes (ORFs: 20, 46, and 177) encoding Xer family recombinases, which are proteins known to participate in resolving the dimeric forms of bacterial chromosomes and circular plasmids [28]. The phage also encodes a Replic_Relax superfamily protein, a ParM-like protein, and a putative ParG-like protein, the common components of plasmid relaxation and partitioning systems. It is known that temperate phages with circular plasmid prophages usually encode Xer recombinases and the components of ParMRC or other segregation systems [29,30,31], which enable extrachromosomal replication in the prophage stage. Therefore, it is possible that Kirov is maintained in the host cytoplasm as a plasmid (plasmid prophage), indicating a temperate lifestyle.

Six protein products (ORFs: 82, 155, 156, 178, 215, and 247) were predicted to contain signal peptides. Signal-peptide-containing proteins are quite common in temperate phages, where they serve various functions, including the regulation of lysis–lysogeny switching [32].

### 2.3. Comparative Genomics

In order to find related phages, a BLASTn search was conducted against the NCBI nr database (taxid:10239) and the *Bacillus* phage database (http://bacillus.phagesdb.org/ assessed on 1 June 2023) using the whole-genome sequence of Kirov as the query sequence. Thirty complete phage genomes, including previously reported phages PBC2 [25], vB_BanS-Tsamsa [27], and Izhevsk [24], were downloaded, and phylogenetic inference was performed with the ViPTree server version 3.6. The resulting phylogram is shown in Figure 4. Moreover, a phylogenetic analysis was conducted using VIRIDIC to assess the relationship between the phage under study and its closest relatives (Appendix A), which supported the data obtained from the VipTree and BLASTn analyses.

Kirov shares 60.2% BLASTn nucleotide identity, 58.4% VIRIDIC intergenomic similarity, and 154 proteins with the *Bacillus*-infecting siphovirus PBC2 (Appendix A), the closest relative found. The linear whole-genome comparison diagram in Figure 5 shows tBLASTx pairwise similarities between Kirov and its closest relatives. Thus, according to the species and genus demarcation criteria [33], Kirov phage can be classified as a new phage species named *Kirovirus kirovense,* specifically referred to as the Kirov strain. *Kirovirus kirovense* Kirov is a member and the founder of the new genus *Kirovirus*, which belongs to the *Andregratiavirinae* subfamily within the *Caudoviricetes* class.

### 2.4. Thermal and pH Stability Assay

The results of the thermal stability assay suggest that the phage is stable for 1 h of incubation at temperatures of 20, 30 and 40 °C, since the phage titers after incubation were similar to the titer of the control sample incubated at 4 °C (Figure 6a). However, at 50 °C, the phage titer decreased by approximately three orders of magnitude compared to the control sample. No phages survived incubation at 60 °C and higher temperatures.

The pH stability test revealed that the phage remains stable within a pH range of 5 to 11 (Figure 6b), as indicated by post-incubation titers comparable to the control (SM+). However, no lytic activity was observed after incubation at pH 2.2, 3, 4, and 12.

### 2.5. Killing Assay

The optical density kinetics of the ATCC 4342 culture at different MOI values showed that within the studied range of multiplicity of infection (MOI) from 0.01 to 10, the phage exhibits lytic properties. The MOI value determines the rate of lysis, so at a MOI equal to 10, the phage took about 1 h to lyse the culture, and at MOI 0.01, it took about 4 h (Figure 7).

### 2.6. Efficiency of Plaque Formation

The efficiency of plaque formation (EOP) was calculated as the ratio of the plaque number observed in the tested strain to the plaque number observed in the ATCC 4342 propagation strain. On the lawns of VKM B-504^T^, VKM B-812, and ATCC 14893 *B. cereus s.l.* strains, the Kirov phage exhibited a 5–10-fold decrease in EOP, while the EOP for *B. cereus* VKM B-373 strain decreased by 50-fold (Figure 8).

### 2.7. Antibacterial Activity of Bacteriophage Kirov in Milk

The composition of a food product may not coincide with the optimal conditions for phages, and some of the product’s components may absorb virions, thereby impairing the bacteriolytic action of phages. To evaluate the effectiveness of the Kirov phage in preventing food contamination, we conducted tests using cow’s milk contaminated with the ATCC 4342 strain at levels of up to 10^5^ CFU/mL.

After 4 h, the ATCC 4342 titer in the sample with the phage decreased below the detection threshold of 10^2^ CFU/mL, while in the control sample, the bacterial titer reached 10^8^ CFU/mL (Figure 9). In samples containing both the phage and cells, the phage titer increased to values as high as 10^8^ PFU/mL.

### 2.8. Viability of B. tropicus ATCC 4342 in Milk Preserved with Kirov Phage

As an additional control method, the viability of bacterial cells in milk was assessed after 6 h of incubation using the rapid resazurin test. In the contaminated milk sample without bacteriophage addition, the resazurin indicator changed color from blue to pink within 15 min due to the action of bacterial-reducing enzymes. On the contrary, in a sample of contaminated milk containing the Kirov phage, the color of the indicator remained unchanged for 30 min, similar to milk samples without bacterial contamination (Figure 10). The results of the experiment indicate a significant difference in the number of active bacterial cells in the milk samples, confirming the effective antibacterial action of Kirov on *B. tropicus* ATCC 4342 cells in milk.

## 3. Discussion

Bacteria belonging to the *Bacillus cereus* group are among the main contaminants of raw milk [34,35,36]. This contamination can occur from the environment, as *B. cereus s.l.* spores and vegetative cells are ubiquitous in nature, or from equipment. While the vegetative cell of *B. cereus* can be inactivated during pasteurization, high-temperature pasteurization can induce spore germination [10]. Next-generation sequencing studies of total bacterial DNA isolated from milk have revealed that up to 5.7% of contaminants correspond to *B. cereus* [34]. In 30 samples obtained from three farms in Poland, the abundance of *Bacilli* DNA ranged from 17.88 to 36.26% [35]. Rubiola et al. demonstrated that the abundance of *Bacilli* in pulled milk samples ranged from 2 to 5%. Additionally, 13 different classes of antibiotic resistance were identified in all milk metagenome samples [36].

Over the past few years, several phages and their enzymes have been described as milk preservatives effective against the most common bacterial contaminants. One such example is LysH5, the endolysin encoded by the staphylococcal bacteriophage phi-SauS-IPLA88. LysH5 was shown to inhibit the growth of *S. aureus* in milk at a concentration of 7.5–15 units/mL (3–6 g/m^3^) [37]. Similarly, the endolysins of streptococcal phages λSA2 and B30 have the ability to reduce streptococci concentrations by 10^2^–10^4^-fold when applied at a concentration of 100 µg/mL (100 g/m^3^) [38]. Guo and coauthors conducted an experiment using an experimental cheese model, where *S. aureus* was co-cultured with recombinant *L. casei* BL23 carrying a plasmid for the expression of Lysdb endolysin from a *Lactobacillus delbrueckii* phage as a secreted protein. The secretion of endolysin phiLdb by *L. casei* BL23 led to a 10^5^-fold decrease in *S. aureus* concentration [39]. Recently, the bacteriophage DLn1 and its endolysin were tested as biocontrol agents against *B. cereus* in milk. DLn1 belongs to the podovirus morphotype, while Kirov belongs to the siphovirus morphotype. DLn1 demonstrated high thermostability and pH stability and was able to reduce the CFU count of *B. cereus* in contaminated milk by up to 10^4^-fold [40]. The bacteriophage Kirov effectively prevents the contamination of milk with ATCC 4342 at a concentration of 10^4^ PFU/mL (Figure 9). Therefore, a volume of several tens of milliliters of a Kirov lysate with a concentration of 10^9^ PFU/mL may be sufficient for preparing a purified biocontrol formulation intended for the preservation of several tons of milk. Thus, the implementation of such a phage in the dairy industry’s safety practices would likely be the most cost-effective. The limited lytic spectrum of Kirov (Figure 8; Appendix A) can be overcome by using it in combination with other phages, such as DLn1 and/or their bacteriolytic enzymes, as a cocktail. The use of phage cocktails has shown promise as an effective approach to broaden the range of targeted bacterial strains and enhance the overall efficacy of phage-based biocontrol strategies in various applications, including food preservation [41,42,43]. In particular, phages have been successfully applied to deactivate and control foodborne pathogens in milk, including *E. coli* [44,45], *Salmonella* spp. [46,47,48], *P. lactis* [49], *S. aureus* [50], *Cronobacter* spp. [51,52], *Y. enterocolitica* [53], and others. Numerous studies have conclusively demonstrated that the application of bacteriophages and phage cocktails serves as a promising biocontrol strategy to safeguard dairy products, particularly milk, from bacterial contamination within the dairy industry.

The Kirov phage shows rapid dynamics of bacterial culture lysis (Figure 7), as well as rapid milk decontamination (Figure 9 and Figure 10). However, the presence of specific temperate phage genes (ORFs: 20, 46, and 177), which could potentially be involved in the formation of lysogen state, requires further detailed study to assess the risk and mechanisms of resistant bacterial strain formation.

In summary, this study presents a novel bacteriophage named *Kirovirus kirovense* Kirov, capable of infecting members of the *Bacillus cereus* group. Through whole-genome sequence comparisons, it has been established that the studied phage is indeed a novel species, officially designated as *Kirovirus kirovense*, with the specific identification of strain Kirov. This phage species belongs to the newly defined genus called *Kirovirus* which falls under the *Andregratiavirinae* subfamily and the *Caudoviricetes* class. Our results show that *Kirovirus kirovense* Kirov has significant potential for inhibiting the growth of *B. cereus s.l*. However, further investigations are required to comprehensively understand the phage’s lifestyle and its interactions with the bacterial population at the genetic level.

## 4. Materials and Methods

### 4.1. Bacterial Strains, Growth Conditions and Reagents

Bacterial strains were obtained from the All-Russian Collection of Microorganisms (VKM), American Type Culture Collection (ATCC), and from other sources. A total of 42 strains were used in this study and are listed in the Appendix A. The bacterial strains were cultivated in Lysogeny broth (LB) and on LB agar (1.5% *w*/*v* and 0.5% *w*/*v* for bottom and top agar layer, respectively) with 10 mM CaCl_2_ and 10 mM MgCl_2_ at 37 °C. All reagents used in this study are listed in Appendix A.

### 4.2. Phage Isolation, Propagation, and Host Range Determination

The phage was isolated from a soil sample collected in Kirov, Kirov region, Russian Federation, and propagated on the sensitive strain *B. tropicus* ATCC 4342. Phage isolation, purification, propagation, and PEG (polyethylene glycol) precipitation were performed as described previously [54]. The resulting phage preparation was formulated in SM+ buffer (50 mM Tris-HCl pH 7.5, 100 mM NaCl, 1 mM MgSO_4_, 0.01% gelatin) and filtered using a sterile filter with a pore size of 0.22 µm. The filtered suspension was subsequently stored at 4 °C for further use. The purified phage preparation was used for host range determination, electron microscopy, and DNA extraction. The host range was determined by the double-layer agar method [55] using 37 strains of the *B. cereus* group as well as 5 other *Bacillus* strains. Briefly, 100 microliters of the phage preparation (at concentrations of 10^4^ and 10^3^ PFU/mL) were combined with 100 µL of bacterial culture (with an optical density at 590 nm (OD590) of 0.8–1.2) and molten LB agar to allow for a final agar concentration of 0.75%. Subsequently, the mixture was gently vortexed for a short period to ensure uniformity. The resulting mixture was poured onto a layer of bottom agar consisting of 1.5% LB agar. Petri dishes were incubated at a temperature of 37 °C overnight.

### 4.3. Transmission Electron Microscopy

For transmission electron microscopy (TEM) analysis, the high-titer phage preparation underwent CsCl density gradient centrifugation. In this procedure, two mL of the phage preparation was meticulously layered onto a preformed CsCl gradient with specific densities of 1.3 g/mL, 1.4 g/mL, 1.5 g/mL, 1.6 g/mL, and 1.7 g/mL, each occupying 2 mL of the gradient. Ultracentrifugation was carried out using a Beckman Coulter L7-55 ultracentrifuge equipped with an SW 40 Ti rotor. The centrifugation process was conducted at a temperature of 18 °C, with a centrifugal force of 110,961.5 g for a duration of 1.5 h.

Following the density gradient centrifugation, five microliters of the phage suspension was loaded onto 400 mesh carbon-formvar-coated copper grids and negatively stained with 1% uranyl acetate. Phage particles were visualized using a JEM-100C transmission electron microscope (JEOL, Akishima, Japan) with an accelerating voltage of 80 kV. Images were taken on Kodak film SO-163 (Kodak, Cat. #74144, Hatfield, PA, USA) with 45,000× magnification. Phage particle dimensions were measured for 10 phage particles using ImageJ version 1.53e [56] in relation to the scale bar generated from the microscope.

### 4.4. Phage DNA Isolation, DNA Sequencing, and Genome Analysis

Phage DNA was extracted using the standard phenol–chloroform extraction protocol described by Sambrook et al. [55] and then sequenced using Illumina with the TruSeq library preparation technology. The genomic sequence was assembled de novo using SPAdes v.3.11.1 software [57]. Open reading frames (ORFs) were identified with RASTtk v.2.0 [58]. The putative functions of the proteins were predicted using BLAST (NCBI) [59] and HHpred [60]. tRNA genes were detected by ARAGORN v1.2.40 [61]. The circular genome visualization was created by BRIG software v.0.95 [62].

### 4.5. Accession Number

The taxon ID of the phage Kirov is 2783539. The phage genome sequence was submitted to GenBank under accession number MW084976. Associated data was submitted under BioProject accession number PRJNA721042, BioSample accession number SAMN18697247 and SRA accession numbers SRR14203650.

### 4.6. Phylogenetic Analysis

In order to find related phages, a BLASTn search [46] was performed against the NCBI GenBank database (taxon: viruses) and the *Bacillus* phages database (BPD) (http://bacillus.phagesdb.org/ assessed 1 June 2023) using the Kirov genome as the query sequence. A linear comparison diagram showing the similarity between phage genomes was visualized with EasyFig v1.4.4 [63]. The number of shared proteins was computed using GET_HOMOLOGUES-est v3.6.1 software [64] with the COGtriangles algorithm [65] (-t 0–C 75). The phylogenomic tree was constructed by the ViPTree server version 3.6 [66]. The intergenomic comparison was performed with the VIRIDIC tool designed for finding intergenomic similarities of prokaryote-infecting viruses [67].

### 4.7. Determination of Packaging Strategy

The phage genome termini were identified by sequencing terminal DNA fragments obtained by the method of rapid amplification of genomic ends (RAGE [68]) as described previously [54,69], with minor modifications. Phage genomic DNA was used for a typical DNA tailing reaction with terminal transferase (New England Biolab, Cat. \# M0315L, Ipswich, MA, USA). Then, two sequential PCR amplifications of both right (R) and left (L) termini were performed using the TaqSE DNA polymerase (SibEnzyme, Cat. #E314, Novosibirsk, Russia) and the appropriate pairs of oligonucleotides provided in the Appendix A. The final PCR products were separated by 1% (*w*/*v*) agarose gel electrophoresis, extracted from the gel, and used for Sanger sequencing with kir1_end2_L 5′-CTGAGTGGTCGGGTTGTAG-3′ and kir1_end5_R 5′-TGGTGTTTATTGCGGTGTTTA-3′ primers for left and right genome termini, respectively.

### 4.8. Thermal and pH Stability Determination

In order to assess the phage stability at various temperatures, aliquots of the phage suspension (in SM+ buffer) with a titer of 6–7 × 10^8^ PFU/mL were incubated at 4, 20, 30, 40, 50, 60, 70, 80, and 90 °C for 1 h. In addition, the phage stability at pH values ranging from 2.2 to 12 was evaluated using four different buffers: glycine–HCl buffer (pH values 2.2 and 3), sodium acetate buffer (pH 4 and 5), phosphate buffer (pH 6, 7 and 8), glycine–NaOH buffer (pH 9 and 10), and Na_2_HPO_4_–NaOH buffer (pH 11–12). The phage suspension was added to each solution to the final concentration of 6–7 × 10^8^ PFU/mL and the mixtures were incubated for 1 h at 37 °C. Aliquots of phage mixed with SM+ buffer were used as controls. The surviving phages from both thermal and pH stability experiments were enumerated by the double-layer agar method. The experiment was performed in five independent trials. The results were visualized in GraphPad Prism 8.4.3 [70,71] as a box-and-whisker diagram, with a 5–95% confidence interval.

### 4.9. Killing Assay

The optical density kinetics of ATCC 4342 cultures at different multiplicity of infection (MOI) values were studied in a 48-well plate. For that, the cultures were cultivated until reaching an OD590 of 0.2, which corresponds to approximately 1 × 10^7^ CFU/mL for this particular strain. Then, 450 µL aliquots of the cultures were added to the plate wells and mixed with 50 µL of the Kirov bacteriophage in SM+ buffer (phage titers: 1 × 10^6^, 1 × 10^7^, 1 × 10^8^, 1 × 10^9^ PFU/mL) to provide a MOI range from 0.01 to 10. Control wells were filled with SM buffer instead of the phage preparation. The plate was incubated at 30 °C in a FilterMax F5 microplate spectrophotometer (Molecular Devices, San Jose, CA, USA), with measurements of OD595 taken every 10 min. The growth curve was plotted using GraphPad Prism 8.4.3 [70,71], and the error bars indicate the standard deviation based on five independent trials.

### 4.10. Efficiency of Plaque Formation on Sensitive B. cereus sensu lato Strains

To evaluate the efficiency of plaque formation on susceptible strains, a phage Kirov dilution sample (10^2^ PFU/mL, enumerated on the host strain ATCC 4342) was used, and the plaques were counted on 12 sensitive strains by the two-layer agar method in 5–6 independent biological trials. The upper agar layer (3 mL) was prepared using 150 μL of overnight bacterial cultures and various volumes of the phage preparation, which were mixed with LB medium so as to equalize the final concentrations of LB-agar to 0.5% and CaCl_2_ and MgCl_2_ salts to 10 mM. Petri dishes were incubated at 37 °C for 22 h before the plaques were counted. The results were depicted as a box-and-whisker plot, showing the 5–95% confidence interval. The visualization was created using GraphPad Prism 8.4.3 [70,71].

### 4.11. Effect of Kirov Phages on Experimentally Contaminated Cow Milk

Ten milliliters of ultra-pasteurized milk (3.2% fat) was artificially contaminated with the ATCC 4342 strain to 10^5^ CFU/mL, followed by the addition of the Kirov phage to a final titer of 10^4^ PFU/mL (MOI = 0.1). In the control sample, SM buffer (routinely used for phage storage) was used instead of the phage preparation. The milk samples were incubated at 37 °C with stirring for 6 h, and bacterial growth was monitored using LB-Miller solid medium every 2 h. The phage titer at the starting point was determined by the double-layer agar method. Serial dilutions of the cultures were prepared using a 0.9% (*w*/*v*) NaCl solution. Experiments were carried out in three independent trials. The results were depicted as a box-and-whisker plot, showing the 5–95% confidence interval. The visualization was created using GraphPad Prism 8.4.3 [70,71].

### 4.12. Resazurin Reduction Test

As an additional control, the activity of bacterial cells in milk was examined after 6 h of incubation using the resazurin reduction test [72]. To perform the test, 0.1 volume of a stock resazurin solution (0.5 mg/mL) was added to the milk, and changes in the solution color were evaluated after 15 and 30 min. The experiments were carried out in three independent trials.

## Figures and Tables

**Figure 1 ijms-24-12584-f001:**
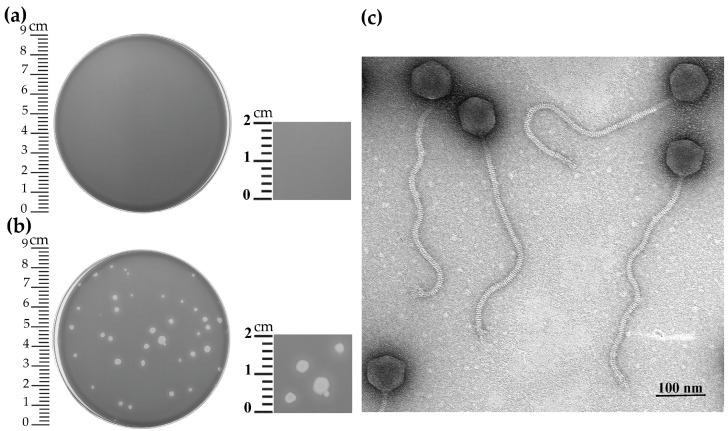
The *Bacillus* phage *Kirovirus kirovense* Kirov plaque morphology and virion morphology: (**a**) control plate (non-infected host strain *B. tropicus* ATCC 4342); (**b**) negative colonies formed by the phage. (**c**) Transmission electron microscopy of the phage particles negatively stained with 1% (*w*/*v*) uranyl acetate. Scale bar represents 100 nm. The full TEM micrograph of *Kirovirus kirovense* Kirov is presented in the Appendix A.

**Figure 2 ijms-24-12584-f002:**
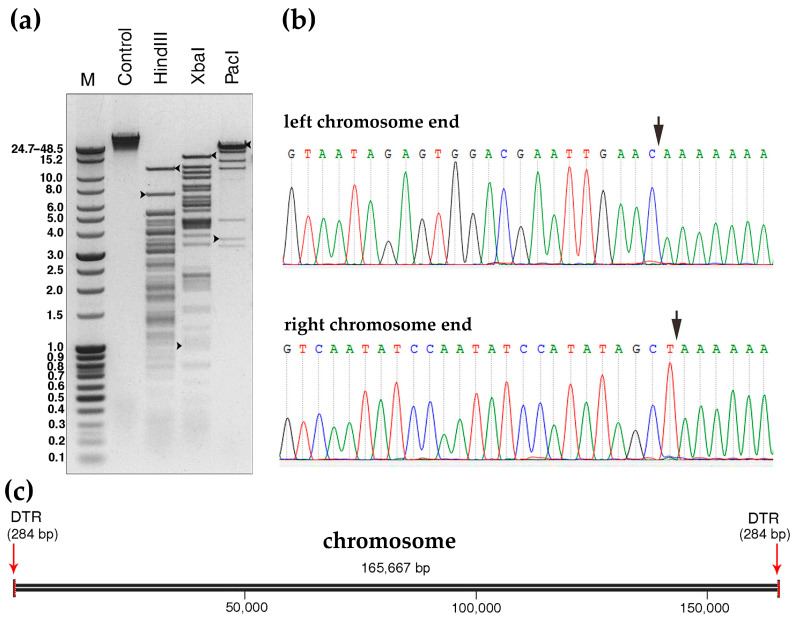
Determination of packaging strategy. (**a**) Restriction analysis of the phage DNA with enzymes HindIII, XbaI, and PacI. M—molecular weight markers, control—intact phage DNA. The restriction fragments containing DTRs are indicated by black arrows. The original full-length gel is presented in Appendix A. (**b**) The terminal sequences of the phage genome determined by RAGE. The terminal regions are indicated by black arrows on the sequencing chromatograms. (**c**) Schematics of *Kirovirus kirovense* Kirov chromosome.

**Figure 3 ijms-24-12584-f003:**
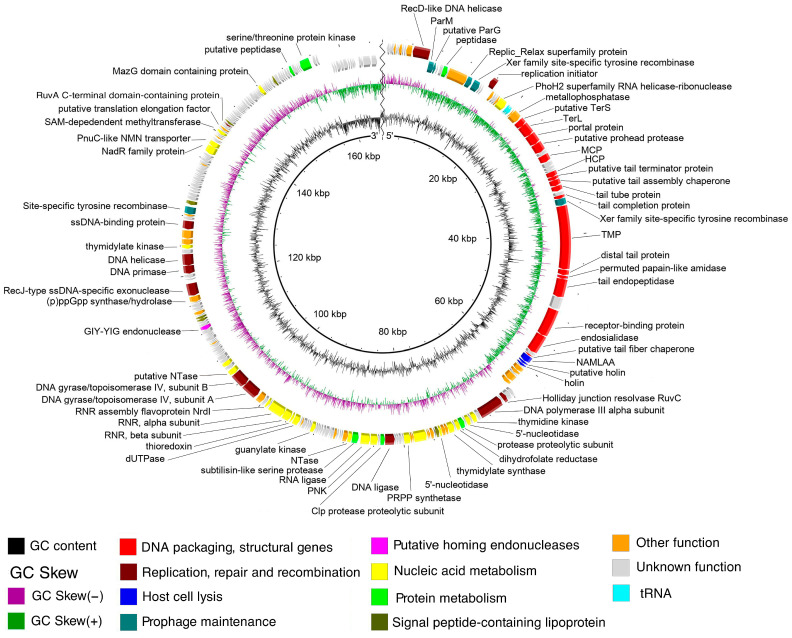
The genome map of the *Kirovirus kirovense* Kirov phage. Functionally assigned ORFs are highlighted based on their general functions (see the legend). Abbreviations used include: TerL—terminase, large subunit; TerS—terminase, small subunit; TCP—tail completion protein; TTP—tail tube protein; TAC—tail assembly chaperone; HCP—head completion protein; MCP—major capsid protein; TMP—tape measure protein; NAMLAA—N-acetylmuramoyl-L-alanine amidase; PRPPP synthetase—phosphoribosylpyrophosphate synthetase; PNK—polynucleotide kinase; NTase—nucleotidyltransferase; RNR—ribonucleotide reductase.

**Figure 4 ijms-24-12584-f004:**
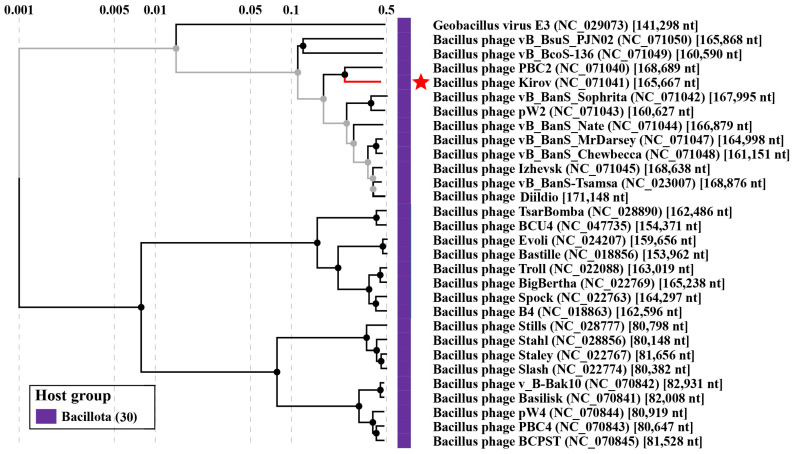
The viral proteomic tree including *Kirovirus kirovense* Kirov and its closest relatives. The *Kirovirus kirovense* Kirov phage is indicated by red star.

**Figure 5 ijms-24-12584-f005:**
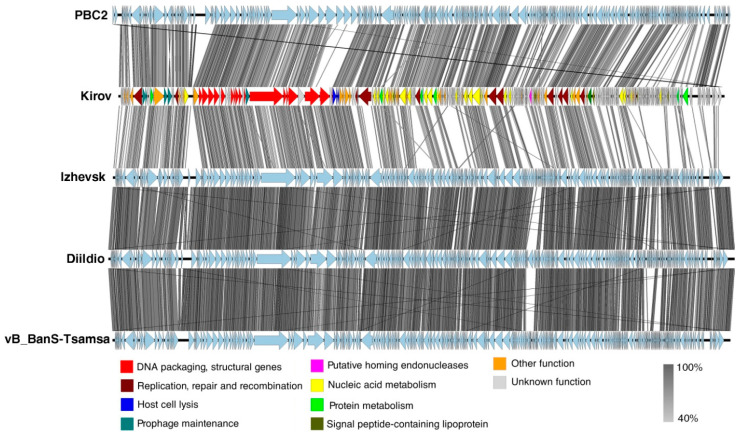
The pairwise tBLASTx whole-genome comparison performed for *Kirovirus kirovense* Kirov and related phages. The Kirov genes are highlighted according to the legend. Gray areas between the genome maps indicate the level of identity (from 40% to 100%, see the legend on the right).

**Figure 6 ijms-24-12584-f006:**
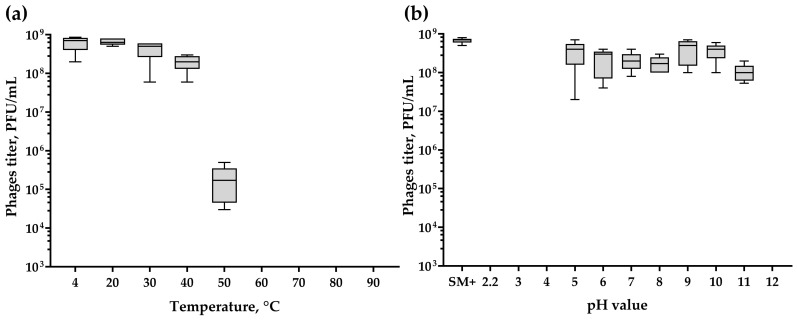
(**a**) Thermal stability and (**b**) pH stability of *Kirovirus kirovense* Kirov phage. The graphs were created in GraphPad Prism 8.4.3 with a 95% confidence interval. Five independent trials were performed for the experiments.

**Figure 7 ijms-24-12584-f007:**
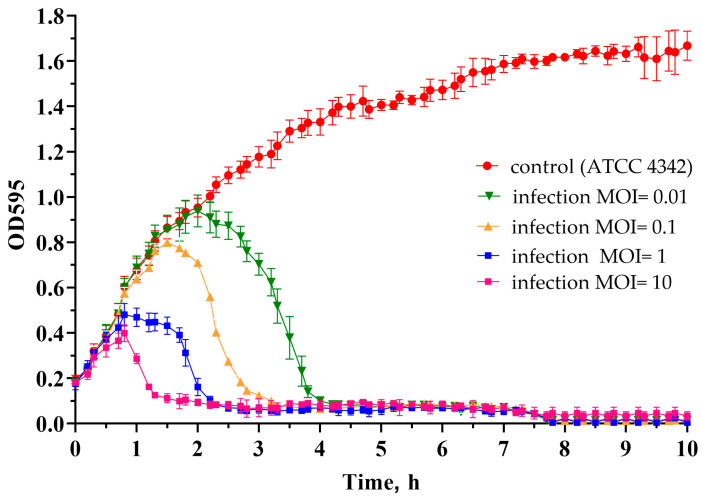
The growth kinetics of *B. tropicus* ATCC 4342 upon infection with *Kirovirus kirovense* Kirov phage at different MOI. A non-infected ATCC 4342 culture was used as a control.

**Figure 8 ijms-24-12584-f008:**
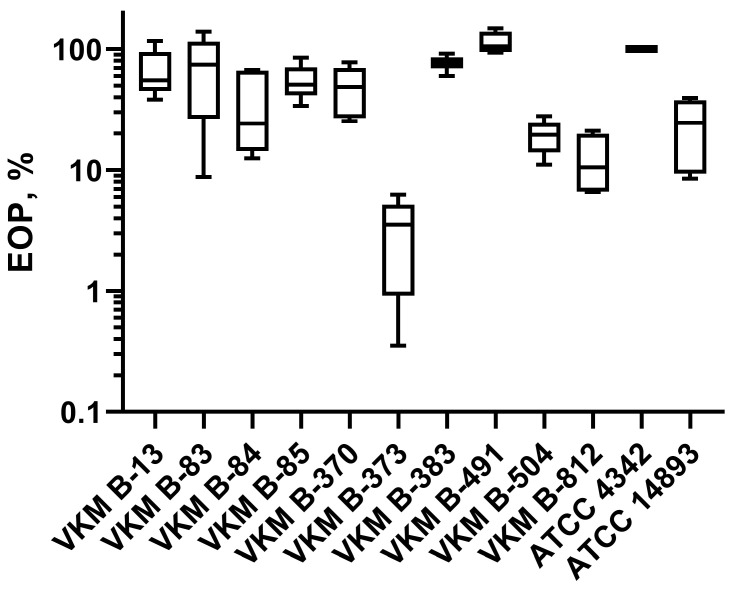
The efficiency of plaque formation of *Kirovirus kirovense* Kirov bacteriophage on the lawns of sensitive *B. cereus s.l.* strains.

**Figure 9 ijms-24-12584-f009:**
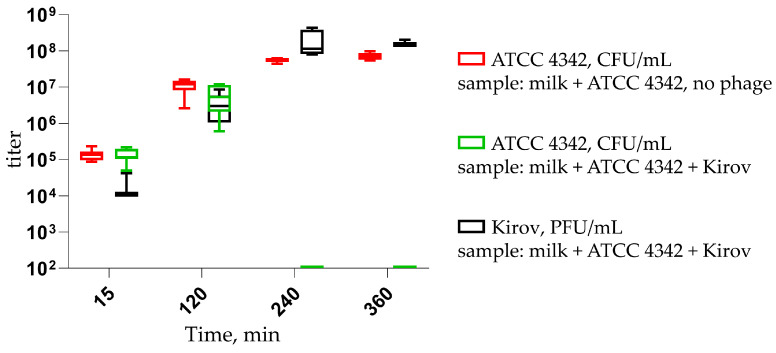
The bacteriolytic effect of phage *Kirovirus kirovense* Kirov in milk experimentally contaminated by *B. tropicus* ATCC 4342 with MOI = 0.1.

**Figure 10 ijms-24-12584-f010:**
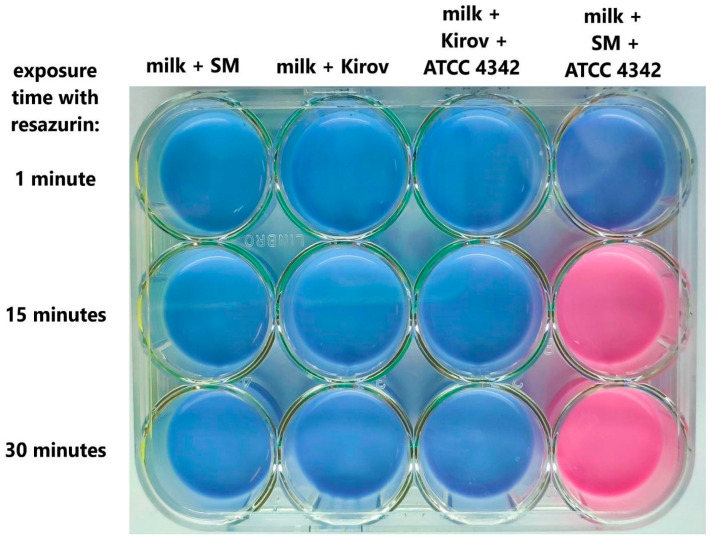
Resazurin test: the viability of bacterial cells in milk artificially contaminated with *B. tropicus* ATCC 4342 was assessed after 6 h of incubation in the presence of the *Kirovirus kirovense* Kirov phage.

## Data Availability

The annotated complete genome of Kirov bacteriophage (*Kirovirus kirovense* Kirov) was deposited into GenBank under accession number MW084976. Associated data were submitted under BioProject accession number PRJNA721042, BioSample accession number SAMN18697247, and SRA accession number SRR14203650.

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
