# Peer review of "A Genomic Analysis of the Bacillus Bacteriophage Kirovirus kirovense Kirov and Its Ability to Preserve Milk"

_ijms, 2023, doi:10.3390/ijms241612584_

Round 1

Reviewer 1 Report

The manuscript by Olesya A. Kazantseva et al. is devoted to the characteristics of a bacteriophage isolated from the soil and designated as Kirovirus kirovense Kirov. The content of the presented MS corresponds to the profile of the IJMS journal. The authors submitted a high-quality manuscript that can be accepted for publication after minor corrections

Line 115 – please, give a link to the article for Izhevsk and PBC2 phages.

Lines 222-225. Information should be moved to Material and Methods section

All Latin names should be stressed by italic (e.g.,  Lines 498, 519)

Author Response

We would like to express our sincere gratitude for your valuable review of our manuscript. We truly appreciate the time and effort you invested in evaluating our work.

Regarding your comments, we have carefully addressed each one to improve the manuscript accordingly. Specifically, in Line 115, we have included the links to the articles on the Izhevsk and PBC2 phages as suggested:

Skorynina, A. V.; Piligrimova, E.G.; Kazantseva, O.A.; Kulyabin, V.A.; Baicher, S.D.; Ryabova, N.A.; Shadrin, A.M. Bacillus-Infecting Bacteriophage Izhevsk Harbors Thermostable Endolysin with Broad Range Specific-ity. PLoS One 2020, 15, doi:10.1371/journal.pone.0242657.

  1. Kong, M.; Na, H.; Ha, N.C.; Ryu, S. LysPBC2, a Novel Endolysin Harboring a Bacillus Cereus Spore Binding Domain. Appl Environ Microbiol 2018, 85, doi:10.1128/AEM.02462-18.

We have removed the redundant information provided in Lines 222-225, as that information is already provided in the Material and Methods section, as per your recommendation.

Unfortunately, we cannot italicize the Latin names in the reference list (as exemplified in Lines 498 and 519) as it is automatically generated and will ultimately be typeset by the journal. Authors are responsible for italicizing Latin names in the main text of the article and supplementary information.

Once again, thank you for your insightful comments and valuable suggestions. Your feedback has undoubtedly enhanced the quality of our manuscript, and we sincerely appreciate your support throughout the review process. All the changes made to the manuscript are marked up using the "Track Changes" feature in the word file of the revised manuscript.

Reviewer 2 Report

The article "The Genomic Analysis of the Bacillus Bacteriophage Kirovirus 2 Kirovense Kirov and its Ability to Preserve Milk" provides a comprehensive exploration of the biology of the Bacillus bacteriophage Kirovirus 2. The authors skillfully describe the intricate details of this bacteriophage, shedding light on its genetic makeup, replication mechanism, and interaction with its bacterial host.

One of the highlights of the article is the demonstration of the practical application of this bacteriophage in preserving milk. The study showcases the potential of using bacteriophages as an alternative to traditional methods for milk preservation. By elaborating on the specific mechanisms through which Kirovirus 2 Kirovense Kirov interacts with harmful bacteria in milk, the researchers present a promising avenue for enhancing the safety and shelf life of dairy products.

Furthermore, the authors advocate the importance of considering the broader context of bacteriophage research, specifically in relation to the effectiveness of bacteriophages as antibacterial agents in milk. They emphasize the need to cite more works that explore the antimicrobial efficiency of bacteriophages in milk preservation. authors should add additional references exploring the phage activity in milk (e.g. https://doi.org/10.1111/jfs.12747). This additional literature strengthens the credibility and relevance of their research, providing a holistic perspective on the potential applications of bacteriophages in the dairy industry.

In conclusion, "The Genomic Analysis of the Bacillus Bacteriophage Kirovirus 2 Kirovense Kirov and its Ability to Preserve Milk" is a well-written and insightful scientific article that contributes significantly to the understanding of bacteriophages and their practical applications in milk preservation. The incorporation of related research on bacteriophage efficacy in milk underscores the broader significance of this work, making it a valuable contribution to the field of microbiology and food science.

Minor editing of the English language required

Author Response

We would like to extend our sincere appreciation for your thoughtful and constructive review of our article titled "The Genomic Analysis of the Bacillus Bacteriophage Kirovirus Kirovense Kirov and its Ability to Preserve Milk."

We are delighted that you found the demonstration of the practical application of Kirovirus Kirovense Kirov in milk preservation to be a highlight of the article. Furthermore, we acknowledge the importance of considering the broader context of bacteriophage research, particularly concerning their effectiveness as antibacterial agents in milk preservation. Your suggestion to include additional references exploring phage activity in milk (e.g., https://doi.org/10.1111/jfs.12747) is valuable, and we have taken it into account.

All the changes made to the manuscript are marked up using the “Track Changes” in the word file of the revised manuscript.

Additionally, we appreciate your comments on the minor editing required for the English language. We will carefully review and revise the manuscript to ensure its clarity and readability.

Once again, thank you for your insightful comments and suggestions. Your feedback has undoubtedly enriched the quality of our article and its potential impact in the fields of microbiology and food science. We are dedicated to ensuring that the revised article meets the highest scientific standards.

Reviewer 3 Report

In this manuscript,Blastn was used to compare nucleotide identity,however,a new method VIRIDIC is much more accurate than Blastn,please refer to"Moraru C, Varsani A, Kropinski AM. VIRIDIC-A Novel Tool to Calculate the Intergenomic Similarities of Prokaryote-Infecting Viruses. Viruses. 2020 Nov 6;12(11):1268. doi: 10.3390/v12111268. PMID: 33172115; PMCID: PMC7694805."

excellent

Author Response

Thank you for your review of our manuscript. We genuinely appreciate your valuable input.

We concur with your suggestion that VIRIDIC, as presented in the work of Moraru et al. ("VIRIDIC-A Novel Tool to Calculate the Intergenomic Similarities of Prokaryote-Infecting Viruses," Viruses, 2020 Nov 6;12(11):1268), offers a more accurate method for estimating the relatedness between more distantly-related phages. However, as stated in the aforementioned article, VIRIDIC is based on the percent identity between two genomes determined by BLASTn, and thus, the sensitivity of VIRIDIC is given by BLASTN. As the difference between the obtained genomic identity data using BLASTn and VIRIDIC is not significant, we kept the previously obtained data from the BLASTn analysis in the table and text, and also included the results from VIRIDIC in the study (see supplementary information).

Moreover, it is worth noting that the article "Turner D, Kropinski AM, Adriaenssens EM. A Roadmap for Genome-Based Phage Taxonomy. Viruses. 2021 Mar 18;13(3):506. doi: 10.3390/v13030506. PMID: 33803862; PMCID: PMC8003253" indicates that several tools, such as BLASTn (% identity multiplied by % coverage), VIRIDIC (intergenomic distance calculator), or CD-HIT-EST, can be used for differentiating between genera and species based on phage genome identity. Therefore, the use of BLASTn is acceptable, although VIRIDIC may be preferable for distantly related phages.

In light of your suggestion, the inclusion of VIRIDIC data has helped to enhance the accuracy of our conclusions and improve the overall scientific rigor of our study.

We are also delighted to receive your positive feedback on the quality of the English language in our manuscript. We strive to maintain a high standard of writing to ensure clarity and coherence in our scientific communication.

Once again, we express our gratitude for your diligent review and helpful recommendations.

All the changes made to the manuscript are marked up using the “Track Changes” in the word file of the revised manuscript.

Reviewer 4 Report

The manuscript " The genomic analysis of the Bacillus bacteriophage Kirovirus 2 kirovense Kirov and its ability to preserve milk" by Kazantseva and co-workers investigated the characteristics of the new bacteriophage Kirovirus kirovense Kirov and its potential to be used against specific milk pathogens. The authors concluded that the application in milk exhibited good results, however more investigations focusing on safe usage of this phage has to be performed in the future. 

The manuscript is suitable for publication in IJMS. 

Author Response

We are immensely grateful for your meticulous review of our manuscript titled "The Genomic Analysis of the Bacillus Bacteriophage Kirovirus Kirovense Kirov and its Ability to Preserve Milk." We are delighted to hear that you found our investigation into the characteristics of the novel bacteriophage Kirovirus kirovense Kirov and its potential application against milk pathogens to be commendable. Your positive evaluation of the manuscript's suitability for publication in IJMS is deeply appreciated. We have dedicated considerable effort to ensure the scientific rigor and clarity of our work, and we are thrilled that it aligns with the standards of the journal.